## PROCEEDINGS A

statistics, cybernetics, biophysics

Markov blanket, variational Bayesian inference, active inference, non-equilibrium steady state, predictive processing, free-energy principle

**Author for correspondence:**
Lancelot Da Costa
e-mail: l.da-costa@imperial.ac.uk

# Bayesian mechanics for stationary processes

Lancelot Da Costa[1,2], Karl Friston[2], Conor Heins[3,4,5] and Grigorios A. Pavliotis[1]

[1]Department of Mathematics, Imperial College London, London SW7 2AZ, UK
[2]Wellcome Centre for Human Neuroimaging, University College London, London WC1N 3AR, UK
[3]Department of Collective Behaviour, Max Planck Institute of Animal Behavior, Konstanz D-78457, Germany
[4]Centre for the Advanced Study of Collective Behaviour, University of Konstanz, Konstanz D-78457, Germany
[5]Department of Biology, University of Konstanz, Konstanz D-78457, Germany

LDC, 0000-0003-0126-4588; KF, 0000-0001-7984-8909

This paper develops a Bayesian mechanics for adaptive systems. Firstly, we model the interface between a system and its environment with a Markov blanket. This affords conditions under which states internal to the blanket encode information about external states. Second, we introduce dynamics and represent adaptive systems as Markov blankets at steady state. This allows us to identify a wide class of systems whose internal states appear to infer external states, consistent with variational inference in Bayesian statistics and theoretical neuroscience. Finally, we partition the blanket into sensory and active states. It follows that active states can be seen as performing active inference and well-known forms of stochastic control (such as PID control), which are prominent formulations of adaptive behaviour in theoretical biology and engineering.

## 1. Introduction

Any object of study must be, implicitly or explicitly, separated from its environment. This implies a boundary that separates it from its surroundings, and which persists for at least as long as the system exists.

In this article, we explore the consequences of a boundary mediating interactions between states internal and external to a system. This provides a useful metaphor to think about biological systems, which comprise spatially bounded, interacting components, nested at several spatial scales [1,2]: for example, the membrane of a cell acts as a boundary through which the cell communicates with its environment, and the same can be said of the sensory receptors and muscles that bound the nervous system.

By examining the dynamics of persistent, bounded systems, we identify a wide class of systems wherein the states internal to a boundary appear to infer those states outside the boundary—a description which we refer to as Bayesian mechanics. Moreover, if we assume that the boundary comprises sensory and active states, we can identify the dynamics of active states with well-known descriptions of adaptive behaviour from theoretical biology and stochastic control.

In what follows, we link a purely mathematical formulation of interfaces and dynamics with descriptions of belief updating and behaviour found in the biological sciences and engineering. Altogether, this can be seen as a model of adaptive agents, as these interface with their environment through sensory and active states and furthermore behave so as to preserve a target steady state.

## (a) Outline of paper

This paper has three parts, each of which introduces a simple, but fundamental, move.

(i) The first is to partition the world into internal and external states whose boundary is modelled with a Markov blanket [3,4]. This allows us to identify conditions under which internal states encode information about external states.
(ii) The second move is to equip this partition with stochastic dynamics. The key consequence of this is that internal states can be seen as continuously inferring external states, consistent with variational inference in Bayesian statistics and with predictive processing accounts of biological neural networks in theoretical neuroscience.
(iii) The third move is to partition the boundary into sensory and active states. It follows that active states can be seen as performing active inference and stochastic control, which are prominent descriptions of adaptive behaviour in biological agents, machine learning and robotics.

## (b) Related work

The emergence and sustaining of complex (dissipative) structures have been subjects of long-standing research starting from the work of Prigogine [5,6], followed notably by Haken's synergetics [7], and in recent years, the statistical physics of adaptation [8]. A central theme of these works is that complex systems can only emerge and sustain themselves far from equilibrium [9–11].

Information processing has long been recognized as a hallmark of cognition in biological systems. In light of this, theoretical physicists have identified basic instances of information processing in systems far from equilibrium using tools from information theory, such as how a drive for metabolic efficiency can lead a system to become predictive [12–15].

A fundamental aspect of biological systems is a self-organization of various interacting components at several spatial scales [1,2]. Much research currently focuses on multipartite processes—modelling interactions between various sub-components that form biological systems—and how their interactions constrain the thermodynamics of the whole [16–20].

At the confluence of these efforts, researchers have sought to explain cognition in biological systems. Since the advent of the twentieth century, Bayesian inference has been used to describe various cognitive processes in the brain [21–25]. In particular, the free energy principle [23], a

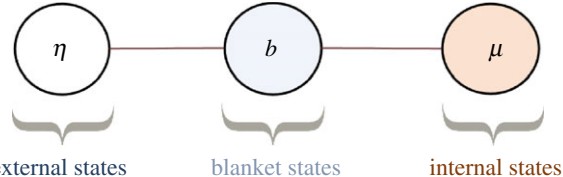

**Figure 1.** Markov blanket depicted graphically as an undirected graphical model, also known as a Markov random field [4,31]. (A Markov random field is a Bayesian network whose directed arrows are replaced by undirected arrows.) The circles represent random variables. The lines represent conditional dependencies between random variables. The Markov blanket condition means that there is no line between $\mu$ and $\eta$. This means that $\mu$ and $\eta$ are conditionally independent given $b$. In other words, knowing the internal state $\mu$, does not afford additional information about the external state $\eta$ when the blanket state $b$ is known. Thus blanket states act as an informational boundary between internal and external states. (Online version in colour.)

prominent theory of self-organization from the neurosciences, postulates that Bayesian inference can be used to describe the dynamics of multipartite, persistent systems modelled as Markov blankets at non-equilibrium steady state [26–30].

This paper connects and develops some of the key themes from this literature. Starting from fundamental considerations about adaptive systems, we develop a physics of things that hold beliefs about other things—consistently with Bayesian inference—and explore how it relates to known descriptions of action and behaviour from the neurosciences and engineering. Our contribution is theoretical: from a biophysicist's perspective, this paper describes how Bayesian descriptions of biological cognition and behaviour can emerge from standard accounts of physics. From an engineer's perspective, this paper contextualizes some of the most common stochastic control methods and reminds us how these can be extended to suit more sophisticated control problems.

## (c) Notation

Let $\Pi \in \mathbb{R}^{d \times d}$ be a square matrix with real coefficients. Let $\eta, b, \mu$ denote a partition of the states $[\![1, d]\!]$, so that

$$\Pi = \begin{bmatrix} \Pi_\eta & \Pi_{\eta b} & \Pi_{\eta \mu} \\ \Pi_{b \eta} & \Pi_b & \Pi_{b \mu} \\ \Pi_{\mu \eta} & \Pi_{\mu b} & \Pi_\mu \end{bmatrix}.$$

We denote principal submatrices with one index only (i.e. we use $\Pi_\eta$ instead of $\Pi_{\eta \eta}$). Similarly, principal submatrices involving various indices are denoted with a colon

$$\Pi_{\eta:b} := \begin{bmatrix} \Pi_\eta & \Pi_{\eta b} \\ \Pi_{b \eta} & \Pi_b \end{bmatrix}.$$

When a square matrix $\Pi$ is symmetric positive-definite we write $\Pi \succ 0$. ker and $\cdot^-$ respectively denote the kernel and Moore–Penrose pseudo-inverse of a linear map or matrix, e.g. a non-necessarily square matrix such as $\Pi_{\mu b}$. In our notation, indexing takes precedence over (pseudo) inversion, for example,

$$\Pi_{\mu b}^- := (\Pi_{\mu b})^- \neq (\Pi^-)_{\mu b}.$$

## 2. Markov blankets

This section formalizes the notion of boundary between a system and its environment as a Markov blanket [3,4], depicted graphically in figure 1. Intuitive examples of a Markov blanket are that of a cell membrane, mediating all interactions between the inside and the outside of the cell, or that of sensory receptors and muscles that bound the nervous system.

To formalize this intuition, we model the world's state as a random variable $x$ with corresponding probability distribution $p$ over a state-space $\mathcal{X} = \mathbb{R}^d$. We partition the state-space of $x$ into *external*, *blanket* and *internal* states:

$$x = (\eta, b, \mu)$$
$$\mathcal{X} = \mathcal{E} \times \mathcal{B} \times \mathcal{I}.$$

External, blanket and internal state-spaces $(\mathcal{E}, \mathcal{B}, \mathcal{I})$ are taken to be Euclidean spaces for simplicity.

A Markov blanket is a statement of conditional independence between internal and external states given blanket states.

**Definition 2.1 (Markov blanket).** A Markov blanket is defined as

$$\eta \perp \mu \mid b. \tag{M.B.}$$

That is, blanket states are a Markov blanket separating $\mu, \eta$ [3,4].

The existence of a Markov blanket can be expressed in several equivalent ways

$$\text{(M.B.)} \iff p(\eta, \mu|b) = p(\eta|b)p(\mu|b) \iff p(\eta|b, \mu) = p(\eta|b) \iff p(\mu|b, \eta) = p(\mu|b). \tag{2.1}$$

For now, we will consider a (non-degenerate) Gaussian distribution $p$ encoding the distribution of states of the world

$$p(x) := \mathcal{N}(x; 0, \Pi^{-1}), \quad \Pi \succ 0,$$

with associated precision (i.e. inverse covariance) matrix $\Pi$. Throughout, we will denote the (positive definite) covariance by $\Sigma := \Pi^{-1}$. Unpacking (2.1) in terms of Gaussian densities, we find that a Markov blanket is equivalent to a sparsity in the precision matrix

$$\text{(M.B.)} \iff \Pi_{\eta\mu} = \Pi_{\mu\eta} = 0. \tag{2.2}$$

**Example 2.2.** For example,

$$\Pi = \begin{bmatrix} 2 & 1 & 0 \\ 1 & 2 & 1 \\ 0 & 1 & 2 \end{bmatrix} \Rightarrow \Sigma_{\eta:b}^{-1} = \begin{bmatrix} 2 & 1 \\ 1 & 1.5 \end{bmatrix}, \Sigma_{b:\mu}^{-1} = \begin{bmatrix} 1.5 & 1 \\ 1 & 2 \end{bmatrix}$$

Then,

$$p(\eta, \mu|b) \propto p(\eta, \mu, b) \propto \exp\left(-\frac{1}{2} x \cdot \Pi x\right)$$

$$\propto \exp\left(-\frac{1}{2} \left[\eta, b\right] \Sigma_{\eta:b}^{-1} \begin{bmatrix} \eta \\ b \end{bmatrix} - \frac{1}{2} \left[b, \mu\right] \Sigma_{b:\mu}^{-1} \begin{bmatrix} b \\ \mu \end{bmatrix}\right) \propto p(\eta, b)p(b, \mu) \propto p(\eta|b)p(\mu|b).$$

Thus, the Markov blanket condition (2.1) holds.

## (a) Expected internal and external states

Blanket states act as an information boundary between external and internal states. Given a blanket state, we can express the conditional probability densities over external and internal states (using (2.1) and [32, proposition 3.13])[1]

$$p(\eta|b) = \mathcal{N}(\eta; \Sigma_{\eta b}\Sigma_b^{-1}b, \Pi_\eta^{-1})$$
$$p(\mu|b) = \mathcal{N}(\mu; \Sigma_{\mu b}\Sigma_b^{-1}b, \Pi_\mu^{-1}). \tag{2.3}$$

---

[1]Note that $\Pi_\eta, \Pi_\mu$ are invertible as principal submatrices of a positive definite matrix.

This enables us to associate with any blanket state its corresponding expected external and expected internal states:

$$\boldsymbol{\eta}(b) = \mathbb{E}[\eta \mid b] = \mathbb{E}_{p(\eta|b)}[\eta] = \Sigma_{\eta b}\Sigma_b^{-1}b \in \mathcal{E}$$

$$\boldsymbol{\mu}(b) = \mathbb{E}[\mu \mid b] = \mathbb{E}_{p(\mu|b)}[\mu] = \Sigma_{\mu b}\Sigma_b^{-1}b \in \mathcal{I}.$$

Pursuing the example of the nervous system, each sensory impression on the retina and oculomotor orientation (blanket state) is associated with an expected scene that caused sensory input (expected external state) and an expected pattern of neural activity in the visual cortex (expected internal state) [33].

## (b) Synchronization map

A central question is whether and how expected internal states encode information about expected external states. For this, we need to characterize a synchronization function $\sigma$, mapping the expected internal state to the expected external state, given a blanket state $\sigma(\boldsymbol{\mu}(b)) = \boldsymbol{\eta}(b)$. This is summarized in the following commutative diagram:

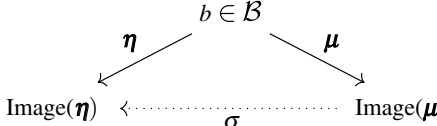

The existence of $\sigma$ is guaranteed, for instance, if the expected internal state completely determines the blanket state—that is, when no information is lost in the mapping $b \mapsto \boldsymbol{\mu}(b)$ in virtue of it being one-to-one. In general, however, many blanket states may correspond to an unique expected internal state. Intuitively, consider the various neural pathways that compress the signal arriving from retinal photoreceptors [34], thus many different (hopefully similar) retinal impressions lead to the same signal arriving in the visual cortex.

### (i) Existence

The key for the existence of a function $\sigma$ mapping expected internal states to expected external states given blanket states, is that for any two blanket states associated with the same expected internal state, these be associated with the same expected external state. This non-degeneracy means that the internal states (e.g. patterns of activity in the visual cortex) have enough capacity to represent all possible expected external states (e.g. three-dimensional scenes of the environment). We formalize this in the following Lemma:

**Lemma 2.3.** *The following are equivalent:*

(i) *There exists a function* $\sigma : \mathrm{Image}(\boldsymbol{\mu}) \to \mathrm{Image}(\boldsymbol{\eta})$ *such that for any blanket state* $b \in \mathcal{B}$

$$\sigma(\boldsymbol{\mu}(b)) = \boldsymbol{\eta}(b).$$

(ii) *For any two blanket states* $b_1, b_2 \in \mathcal{B}$

$$\boldsymbol{\mu}(b_1) = \boldsymbol{\mu}(b_2) \Rightarrow \boldsymbol{\eta}(b_1) = \boldsymbol{\eta}(b_2).$$

(iii) $\ker \Sigma_{\mu b} \subset \ker \Sigma_{\eta b}$.
(iv) $\ker \Pi_{\mu b} \subset \ker \Pi_{\eta b}$.

See appendix A for a proof of lemma 2.3.

**Example 2.4.**

— When external, blanket and internal states are one dimensional, the existence of a synchronization map is equivalent to $\Pi_{\mu b} \neq 0$ or $\Pi_{\mu b} = \Pi_{\eta b} = 0$.

— If $\Pi_{\mu b}$ is chosen at random—its entries sampled from a non-degenerate Gaussian or uniform distribution—then $\Pi_{\mu b}$ has full rank with probability 1. If furthermore, the blanket state-space $\mathcal{B}$ has lower or equal dimensionality than the internal state-space $\mathcal{I}$, we obtain that $\Pi_{\mu b}$ is one-to-one (i.e. $\ker \Pi_{\mu b} = 0$) with probability 1. Thus, in this case, the conditions of lemma 2.3 are fulfilled with probability 1.

## (ii) Construction

The key idea to map an expected internal state $\mu(b)$ to an expected external state $\eta(b)$ is to: (1) find a blanket state that maps to this expected internal state (i.e. by inverting $\mu$) and (2) from this blanket state, find the corresponding expected external state (i.e. by applying $\eta$):

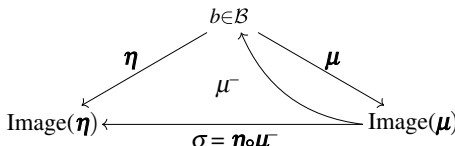

We now proceed to solving this problem. Given an internal state $\mu$, we study the set of blanket states $b$ such that $\mu(b) = \mu$

$$\mu(b) = \Sigma_{\mu b} \Sigma_b^{-1} b = \mu \iff b \in \mu^{-1}(\mu) = \Sigma_b \Sigma_{\mu b}^{-1} \mu. \tag{2.4}$$

Here, the inverse on the right-hand side of (2.4) is understood as the preimage of a linear map. We know that this system of linear equations has a vector space of solutions given by [35]

$$\mu^{-1}(\mu) = \{\Sigma_b \Sigma_{\mu b}^- \mu + (\mathrm{Id} - \Sigma_b \Sigma_{\mu b}^- \Sigma_{\mu b} \Sigma_b^{-1}) b : b \in \mathcal{B}\}. \tag{2.5}$$

Among these, we choose

$$\mu^-(\mu) = \Sigma_b \Sigma_{\mu b}^- \mu.$$

**Definition 2.5 (Synchronization map).** We define a synchronization function that maps to an internal state a corresponding most likely internal state[2,3]

$$\sigma : \mathrm{Image}\,\mu \to \mathrm{Image}\,\eta$$

$$\mu \mapsto \eta(\mu^-(\mu)) = \Sigma_{\eta b} \Sigma_{\mu b}^- \mu = \Pi_\eta^{-1} \Pi_{\eta b} \Pi_{\mu b}^- \Pi_\mu \mu.$$

The expression in terms of the precision matrix is a by-product of appendix A.

Note that we can always define such $\sigma$, however, it is only when the conditions of lemma 2.3 are fulfilled that $\sigma$ maps expected internal states to expected external states $\sigma(\mu(b)) = \eta(b)$. When this is not the case, the internal states do not fully represent external states, which leads to a partly degenerate type of representation, see figure 2 for a numerical illustration obtained by sampling from a Gaussian distribution, in the non-degenerate ($a$) and degenerate cases ($b$), respectively.

# 3. Bayesian mechanics

In order to study the time-evolution of systems with a Markov blanket, we introduce dynamics into the external, blanket and internal states. Henceforth, we assume a synchronization map under the conditions of lemma 2.3.

---

[2]This mapping was derived independently of our work in [36, §3.2].

[3]Replacing $\mu^-(\mu)$ by any other element of (2.5) would lead to the same synchronization map provided that the conditions of lemma 2.3 are satisfied.

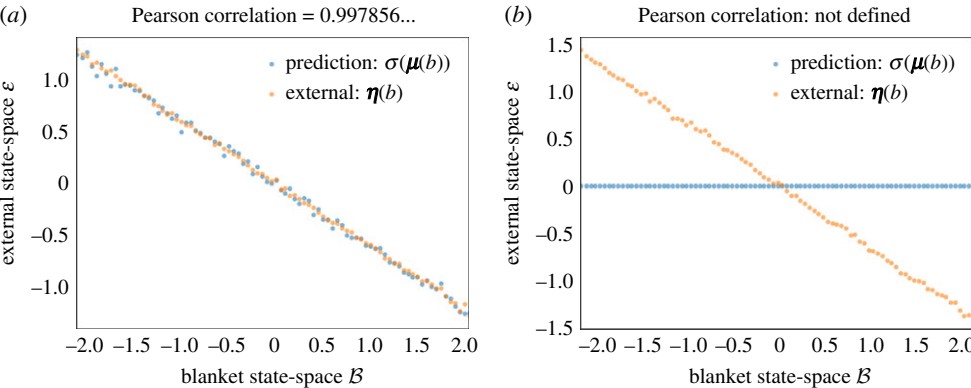

**Figure 2.** Synchronization map: example and non-example. This figure plots expected external states given blanket states $\eta(b)$ (in orange), and the corresponding prediction encoded by internal states $\sigma(\mu(b))$ (in blue). In this example, external, blanket and internal state-spaces are taken to be one dimensional. We show the correspondence when the conditions of lemma 2.3 are satisfied (a) and when these are not satisfied (b). In the latter case, the predictions are uniformly zero. To generate these data, (1) we drew $10^6$ samples from a Gaussian distribution with a Markov blanket, (2) we partitioned the blanket state-space into several bins, (3) we obtained the expected external and internal states given blanket states empirically by averaging samples from each bin, and finally, (4) we applied the synchronization map to the (empirical) expected internal states given blanket states. (Online version in colour.)

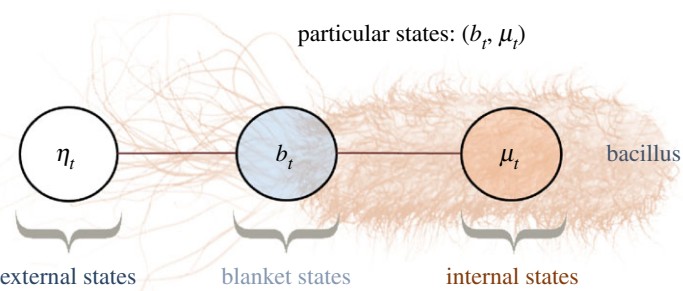

**Figure 3.** Markov blanket evolving in time. We use a bacillus to depict an intuitive example of a Markov blanket that persists over time. Here, the blanket states represent the membrane and actin filaments of the cytoskeleton, which mediate all interactions between internal states and the external medium (external states). (Online version in colour.)

## (a) Processes at a Gaussian steady state

We consider stochastic processes at a Gaussian steady-state $p$ with a Markov blanket. The steady-state assumption means that the system's overall configuration persists over time (e.g. it does not dissipate). In other words, we have a Gaussian density $p = \mathcal{N}(0, \Pi^{-1})$ with a Markov blanket (2.2) and a stochastic process distributed according to $p$ at every point in time

$$x_t \sim p = \mathcal{N}(0, \Pi^{-1}) \quad \text{for any } t.$$

Recalling our partition into external, blanket and internal states, this affords a Markov blanket that persists over time, see figure 3

$$x_t = (\eta_t, b_t, \mu_t) \sim p \Rightarrow \eta_t \perp \mu_t \mid b_t. \tag{3.1}$$

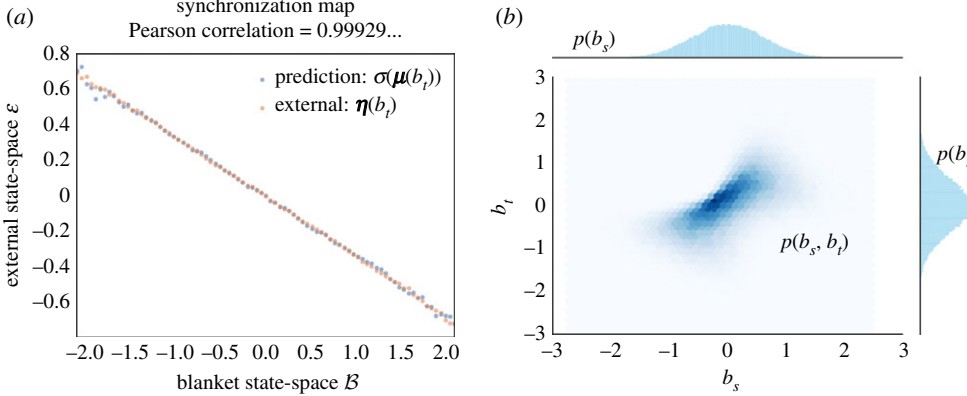

**Figure 4.** Processes at a Gaussian steady state. This figure illustrates the synchronization map and transition probabilities of processes at a Gaussian steady state. (*a*) We plot the synchronization map as in figure 2, only, here, the samples are drawn from trajectories of a diffusion process (3.2) with a Markov blanket. Although this is not the case here, one might obtain a slightly noisier correspondence between predictions $\sigma(\boldsymbol{\mu}(b_t))$ and expected external states $\boldsymbol{\eta}(b_t)$—compared to figure 2—in numerical discretizations of a diffusion process. This is because the steady state of a numerical discretization usually differs slightly from the steady state of the continuous-time process [37]. (*b*) This panel plots the transition probabilities of the same diffusion process (3.2), for the blanket state at two different times. The joint distribution (depicted as a heat map) is not Gaussian but its marginals—the steady-state density—are Gaussian. This shows that in general, processes at a Gaussian steady state are not Gaussian processes. In fact, the Ornstein–Uhlenbeck process is the only stationary diffusion process (3.2) that is a Gaussian process, so the transition probabilities of nonlinear diffusion processes (3.2) are never multivariate Gaussians. (Online version in colour.)

Note that we do not require $x_t$ to be independent samples from the steady-state distribution $p$. On the contrary, $x_t$ may be generated by extremely complex, nonlinear, and possibly stochastic equations of motion. See example 3.1 and figure 4 for details.

**Example 3.1.** The dynamics of $x_t$ are described by a stochastic process at a Gaussian steady-state $p = \mathcal{N}(0, \Pi^{-1})$. There is a large class of such processes, which includes:

— Stationary diffusion processes, with initial condition $x_0 \sim p$. Their time-evolution is given by an Itô stochastic differential equation (see appendix B):

$$dx_t = (\Gamma + Q)(x_t)\nabla \log p(x_t)dt + \nabla \cdot (\Gamma + Q)(x_t)dt + \varsigma(x_t)dW_t,$$
$$= -(\Gamma + Q)(x_t)\Pi x_t dt + \nabla \cdot (\Gamma + Q)(x_t)dt + \varsigma(x_t)dW_t \tag{3.2}$$
$$\Gamma := \varsigma\varsigma^{\top}/2, \quad Q = -Q^{\top}.$$

Here, $W_t$ is a standard Brownian motion (a.k.a., Wiener process) [38,39] and $\varsigma, \Gamma, Q$ are sufficiently well-behaved matrix fields (see appendix B). Namely, $\Gamma$ is the diffusion tensor (half the covariance of random fluctuations), which drives dissipative flow; $Q$ is an arbitrary antisymmetric matrix field which drives conservative (i.e. solenoidal) flow. Note that there are no non-degeneracy conditions on the matrix field $\varsigma$—in particular, the process is allowed to be non-ergodic or even completely deterministic (i.e. $\varsigma \equiv 0$). Also, $\nabla \cdot$ denotes the divergence of a matrix field defined as $(\nabla \cdot (\Gamma + Q))_i := \sum_j (\partial/\partial x_j)(\Gamma + Q)_{ij}$.

— More generally, $x_t$ could be generated by any Markov process at steady-state $p$, such as the zig-zag process or the bouncy particle sampler [40–42], by any mean-zero Gaussian process at steady-state $p$ [43], or by any random dynamical system at steady-state $p$ [44].

**Remark 3.2.** When the dynamics are given by an Itô stochastic differential equation (3.2), a Markov blanket of the steady-state density (2.2) does not preclude reciprocal influences between

internal and external states [45,46]. For example,

$$\Pi = \begin{bmatrix} 2 & 1 & 0 \\ 1 & 2 & 1 \\ 0 & 1 & 2 \end{bmatrix}, \quad Q \equiv \begin{bmatrix} 0 & 0 & 1 \\ 0 & 0 & 0 \\ -1 & 0 & 0 \end{bmatrix}, \quad \varsigma \equiv \mathrm{Id}_3$$

and

$$\Rightarrow d \begin{bmatrix} \eta_t \\ b_t \\ \mu_t \end{bmatrix} = - \begin{bmatrix} 1 & 1.5 & 2 \\ 0.5 & 1 & 0.5 \\ -2 & -0.5 & 1 \end{bmatrix} \begin{bmatrix} \eta_t \\ b_t \\ \mu_t \end{bmatrix} dt + \varsigma dW_t.$$

Conversely, the absence of reciprocal coupling between two states in the drift in some instances, though not always, leads to conditional independence [30,36,45].

## (b) Maximum a posteriori estimation

The Markov blanket (3.1) allows us to exploit the construction of §2 to determine expected external and internal states given blanket states

$$\eta_t := \eta(b_t) \quad \mu_t := \mu(b_t).$$

Note that $\eta, \mu$ are linear functions of blanket states; since $b_t$ generally exhibits rough sample paths, $\eta_t, \mu_t$ will also exhibit very rough sample paths.

We can view the steady-state density $p$ as specifying the relationship between external states ($\eta$, causes) and particular states ($b, \mu$, consequences). In statistics, this corresponds to a generative model, a probabilistic specification of how (external) causes generate (particular) consequences.

By construction, the expected internal states encode expected external states via the synchronization map

$$\sigma(\mu_t) = \eta_t,$$

which manifests a form of generalized synchrony across the Markov blanket [47–49]. Moreover, the expected internal state $\mu_t$ effectively follows the most likely cause of its sensations

$$\sigma(\mu_t) = \arg\max p(\eta_t \mid b_t) \quad \text{for any } t.$$

This has an interesting statistical interpretation as expected internal states perform maximum *a posteriori* (MAP) inference over external states.

## (c) Predictive processing

We can go further and associate with each internal state $\mu$ a probability distribution over external states, such that each internal state encodes beliefs about external states

$$q_\mu(\eta) := \mathcal{N}(\eta; \sigma(\mu), \Pi_\eta^{-1}). \tag{3.3}$$

We will call $q_\mu$ the approximate posterior belief associated with the internal state $\mu$ due to the forecoming connection to inference. Under this specification, the mean of the approximate posterior depends upon the internal state, while its covariance equals that of the true posterior w.r.t. external states (2.3). It follows that the approximate posterior equals the true posterior when the internal state $\mu$ equals the expected internal state $\mu(b)$ (given blanket states):

$$q_\mu(\eta) = p(\eta|b) \iff \mu = \mu(b). \tag{3.4}$$

Note a potential connection with epistemic accounts of quantum mechanics; namely, a world governed by classical mechanics ($\sigma \equiv 0$ in (3.2)) in which each agent encodes Gaussian beliefs about external states could appear to the agents as reproducing many features of quantum mechanics [50].

Under this specification (3.4), expected internal states are the unique minimizer of a Kullback–Leibler divergence [51]

$$\boldsymbol{\mu}_t = \arg\min_\mu D_{KL}[q_\mu(\eta)||p(\eta|b)],$$

that measures the discrepancy between beliefs about the external world $q_\mu(\eta)$ and the posterior distribution over external variables. Computing the KL divergence (see appendix C), we obtain

$$\boldsymbol{\mu}_t = \arg\min_\mu (\sigma(\mu) - \boldsymbol{\eta}_t) \Pi_\eta (\sigma(\mu) - \boldsymbol{\eta}_t). \tag{3.5}$$

In the neurosciences, the right-hand side of (3.5) is commonly known as a (squared) precision-weighted prediction error: the discrepancy between the prediction and the (expected) state of the environment is weighted with a precision matrix [24,52,53] that derives from the steady-state density. This equation is formally similar to that found in predictive coding formulations of biological function [24,54–56], which stipulate that organisms minimize prediction errors, and in doing so optimize their beliefs to match the distribution of external states.

## (d) Variational Bayesian inference

We can go further and associate expected internal states to the solution to the classical variational inference problem from statistical machine learning [57] and theoretical neurobiology [52,58]. Expected internal states are the unique minimizer of a free energy functional (i.e. an evidence bound [57,59])

$$F(b_t, \mu_t) \geq F(b_t, \boldsymbol{\mu}_t)$$

$$F(b, \mu) := D_{KL}[q_\mu(\eta)||p(\eta|b)] - \log p(b, \mu)$$

$$= \underbrace{\mathbb{E}_{q_\mu(\eta)}[-\log p(x)]}_{\text{Energy}} - \underbrace{H[q_\mu]}_{\text{Entropy}}. \tag{3.6}$$

The last line expresses the free energy as a difference between energy and entropy: energy or accuracy measures to what extent predicted external states are close to the true external states, while entropy penalizes beliefs that are overly precise.

At first sight, variational inference and predictive processing are solely useful to characterize the average internal state given blanket states at steady state. It is then surprising to see that the free energy says a great deal about a system's expected trajectories as it relaxes to steady state. Figures 5 and 6 illustrate the time-evolution of the free energy and prediction errors after exposure to a surprising stimulus. In particular, figure 5 averages internal variables for any blanket state: In the neurosciences, perhaps the closest analogy is the event-triggered averaging protocol, where neurophysiological responses are averaged following a fixed perturbation, such a predictable neural input or an experimentally controlled sensory stimulus (e.g. spike-triggered averaging, event-related potentials) [62–64].

The most striking observation is the nearly monotonic decrease of the free energy as the system relaxes to steady state. This simply follows from the fact that regions of high density under the steady-state distribution have a low free energy. This *overall decrease* in free energy is the essence of the free-energy principle, which describes self-organization at non-equilibrium steady state [23,28,29]. Note that the free energy, even after averaging internal variables, may decrease non-monotonically. See the explanation in figure 5.

## 4. Active inference and stochastic control

In order to model agents that interact with their environment, we now partition blanket states into sensory and active states

$$b_t = (s_t, a_t)$$

$$x_t = (\eta_t, s_t, a_t, \mu_t).$$

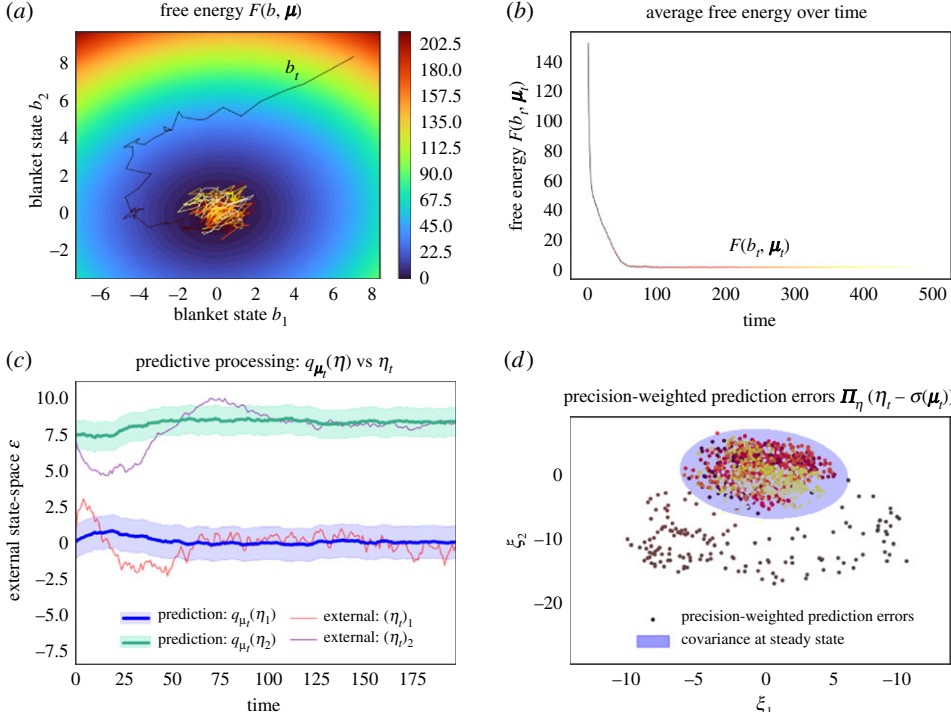

**Figure 5.** Variational inference and predictive processing, averaging internal variables for any blanket state. This figure illustrates a system's behaviour after experiencing a surprising blanket state, averaging internal variables for any blanket state. This is a multidimensional Ornstein–Uhlenbeck process, with two external, blanket and internal variables, initialized at the steady-state density conditioned upon an improbable blanket state $p(x_0|b_0)$. (a) We plot a sample trajectory of the blanket states as these relax to steady state over a contour plot of the free energy (up to a constant). (b) This plots the free energy (up to a constant) over time, averaged over multiple trajectories. In this example, the rare fluctuations that climb the free energy landscape vanish on average, so that the average free energy decreases monotonically. This need not always be the case: conservative systems (i.e. $\varsigma \equiv 0$ in (3.2)) are deterministic flows along the contours of the steady-state density (see appendix B). Since these contours do not generally coincide with those of $F(b, \boldsymbol{\mu})$ it follows that the free energy oscillates between its maximum and minimum value over the system's periodic trajectory. Luckily, conservative systems are not representative of dissipative, living systems. Yet, it follows that the average free energy of expected internal variables may increase, albeit only momentarily, in dissipative systems (3.2) whose solenoidal flow dominates dissipative flow. (c) We illustrate the accuracy of predictions over external states of the sample path from a. At steady state (from timestep $\sim 100$), the predictions become accurate. The prediction of the second component is offset by four units for greater visibility, as can be seen from the longtime behaviour converging to four instead of zero. (d) We show how precision-weighted prediction errors $\xi := \boldsymbol{\Pi}_\eta(\eta_t - \sigma(\boldsymbol{\mu}_t))$ evolve over time. These become normally distributed with zero mean as the process reaches steady state. (Online version in colour.)

Intuitively, sensory states are the sensory receptors of the system (e.g. olfactory or visual receptors) while active states correspond to actuators through which the system influences the environment (e.g. muscles). See figure 7. The goal of this section is to explain how autonomous states (i.e. active and internal states) respond adaptively to sensory perturbations in order to maintain the steady state, which we interpret as the agent's preferences or goal. This allows us to relate the dynamics of autonomous states to active inference and stochastic control, which are well-known formulations of adaptive behaviour in theoretical biology and engineering.

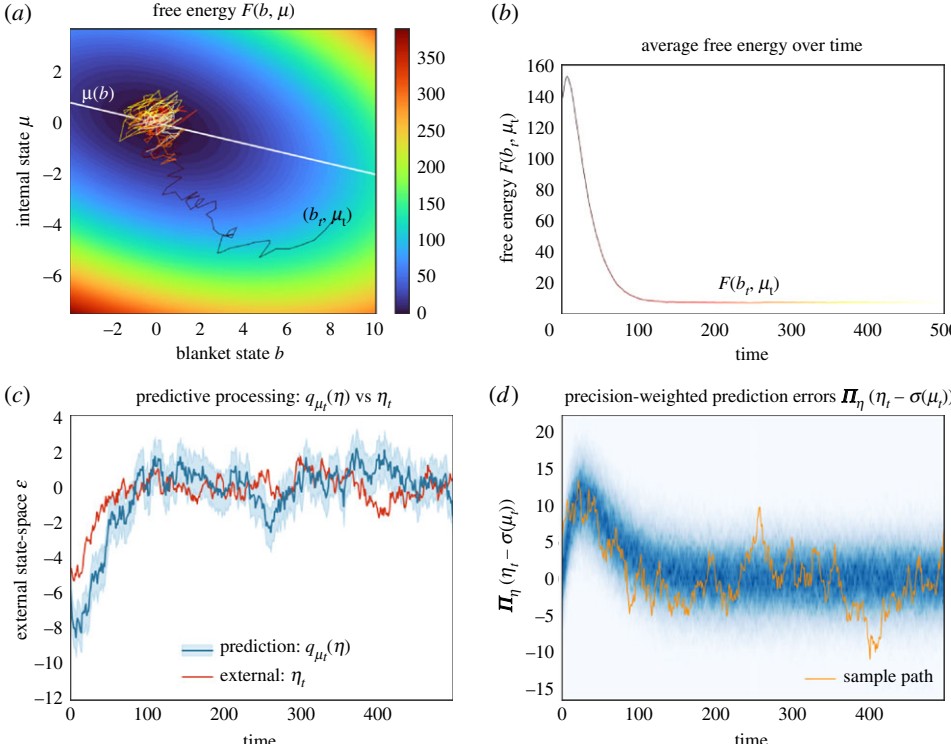

**Figure 6.** Variational inference and predictive processing. This figure illustrates a system's behaviour after experiencing a surprising blanket state. This is a multidimensional Ornstein–Uhlenbeck process, with one external, blanket and internal variable, initialized at the steady-state density conditioned upon an improbable blanket state $p(x_0|b_0)$. (*a*) This plots a sample trajectory of particular states as these relax to steady state over a contour plot of the free energy. The white line shows the expected internal state given blanket states, at which point inference is exact. After starting close to this line, the process is driven by solenoidal flow to regions where inference is inaccurate. Yet, solenoidal flow makes the system converge faster to steady state [60,61] at which point inference becomes accurate again. (*b*) This plots the free energy (up to a constant) over time, averaged over multiple trajectories. (*c*) We illustrate the accuracy of predictions over external states of the sample path from the upper left panel. These predictions are accurate at steady state (from timestep $\sim 100$). (*d*) We illustrate the (precision weighted) prediction errors over time. In orange, we plot the prediction error corresponding to the sample path in *a*; the other sample paths are summarized as a heat map in blue. (Online version in colour.)

## (a) Active inference

We now proceed to characterize autonomous states, given sensory states, using the free energy. Unpacking blanket states, the free energy (3.6) reads

$$F(s, a, \mu) = D_{KL}[q_\mu(\eta) || p(\eta|s, a)] - \log p(\mu|s, a) - \log p(a|s) - \log p(s).$$

Crucially, it follows that the expected autonomous states minimize free energy

$$F(s_t, a_t, \mu_t) \geq F(s_t, \boldsymbol{a}_t, \boldsymbol{\mu}_t)$$

$$\boldsymbol{a}_t := \boldsymbol{a}(s_t) := \mathbb{E}_{p(a_t|s_t)}[a_t] = \Sigma_{as}\Sigma_s^{-1}s_t,$$

where $\boldsymbol{a}_t$ denotes the expected active states given sensory states, which is the mean of $p(a_t|s_t)$. This result forms the basis of active inference, a well-known framework to describe and generate adaptive behaviour in neuroscience, machine learning and robotics [25,58,65–72]. See figure 8.

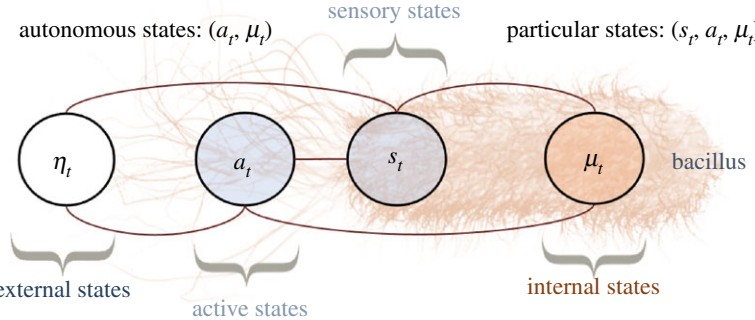

**Figure 7.** Markov blanket evolving in time comprising sensory and active states. We continue the intuitive example from figure 3 of the bacillus as representing a Markov blanket that persists over time. The only difference is that we partition blanket states into sensory and active states. In this example, the sensory states can be seen as the bacillus' membrane, while the active states correspond to the actin filaments of the cytoskeleton.

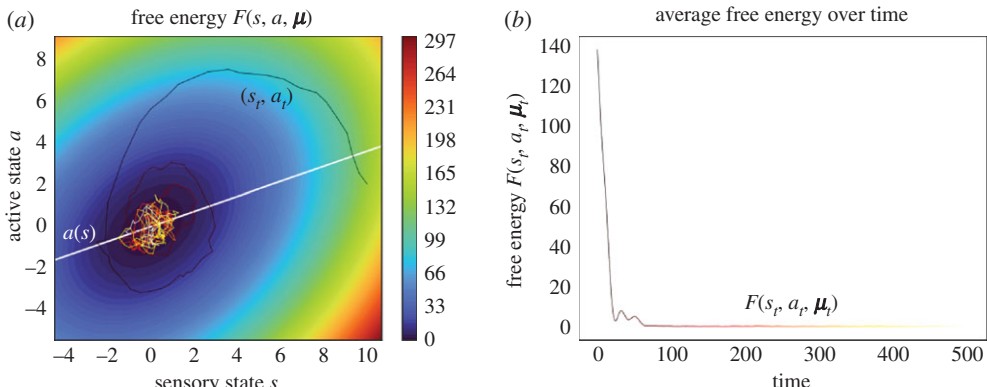

**Figure 8.** Active inference. This figure illustrates a system's behaviour after experiencing a surprising sensory state, averaging internal variables for any blanket state. We simulated an Ornstein–Uhlenbeck process with two external, one sensory, one active and two internal variables, initialized at the steady-state density conditioned upon an improbable sensory state $p(x_0 \mid s_0)$. ($a$) The white line shows the expected active state given sensory states: this is the action that performs active inference and optimal stochastic control. As the process experiences a surprising sensory state, it initially relaxes to steady state in a winding manner due to the presence of solenoidal flow. Even though solenoidal flow drives the actions away from the optimal action initially, it allows the process to converge faster to steady state [60,61,73] where the actions are again close to the optimal action from optimal control. ($b$) We plot the free energy of the expected internal state, averaged over multiple trajectories. In this example, the average free energy does not decrease monotonically—see figure 5 for an explanation. (Online version in colour.)

## (b) Multivariate control

Active inference is used in various domains to simulate control [65,69,71,72,74–77], thus, it is natural that we can relate the dynamics of active states to well-known forms of stochastic control.

By computing the free energy explicitly (see appendix C), we obtain that

$$(a_t, \mu_t) \quad \text{minimizes} \quad (a, \mu) \mapsto \begin{bmatrix} s_t, a, \mu \end{bmatrix} K \begin{bmatrix} s_t \\ a \\ \mu \end{bmatrix} \tag{4.1}$$

$$K := \Sigma_{b:\mu'}^{-1},$$

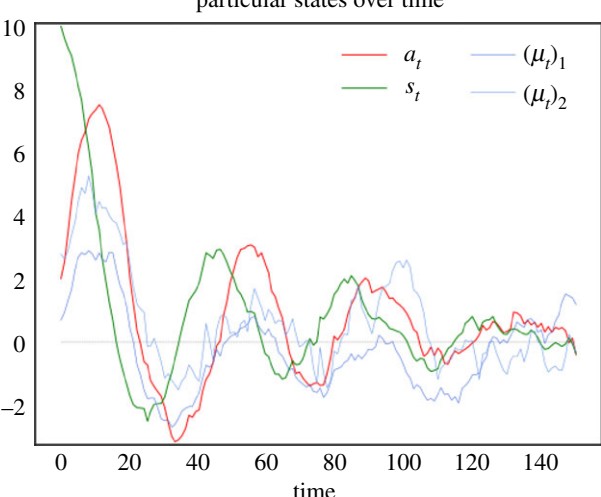

**Figure 9.** Stochastic control. This figure plots a sample path of the system's particular states after it experiences a surprising sensory state. This is the same sample path as shown in figure 8*a*; however, here the link with stochastic control is easier to see. Indeed, it looks as if active states (in red) are actively compensating for sensory states (in green): rises in active states lead to plunges in sensory states and vice versa. Note the initial rise in active states to compensate for the initial perturbation in the sensory states. Furthermore, active states follow a similar trajectory as sensory states, with a slight delay, which can be interpreted as a reaction time [78]. In fact, the correspondence between sensory and active states is a consequence of the solenoidal flow—see figure 8*a*. The damped oscillations as the particular states approach their target value of 0 (in grey) is analogous to that found in basic implementations of stochastic control, e.g. [79, fig. 4.9]. (Online version in colour.)

where we denoted by $K$ the concentration (i.e. precision) matrix of $p(s,a,\mu)$. We may interpret $(a,\mu)$ as controlling how far particular states $[s,a,\mu]$ are from their target set-point of $[0,0,0]$, where the error is weighted by the precision matrix $K$. See figure 9. (Note that we could choose any other set-point by translating the frame of reference or equivalently choosing a Gaussian steady-state centred away from zero). In other words, there is a cost associated with how far away $s,a,\mu$ are from the origin and this cost is weighed by the precision matrix, which derives from the stationary covariance of the steady state. In summary, the expected internal and active states can be seen as performing multivariate stochastic control, where the matrix $K$ encodes control gains. From a biologist's perspective, this corresponds to a simple instance of homeostatic regulation: maintaining physiological variables within their preferred range.

## (c) Stochastic control in an extended state-space

More sophisticated control methods, such as PID (proportional-integral-derivative) control [77,80], involve controlling a process and its higher orders of motion (e.g. integral or derivative terms). So how can we relate the dynamics of autonomous states to these more sophisticated control methods? The basic idea involves extending the sensory state-space to replace the sensory process $s_t$ by its various orders of motion $\tilde{s}_t = (s_t^{(0)}, \ldots, s_t^{(n)})$ (integral, position, velocity, jerk etc, up to order $n$). To find these orders of motion, one must solve the stochastic realization problem.

### (i) The stochastic realization problem

Recall that the sensory process $s_t$ is a stationary stochastic process (with a Gaussian steady state). The following is a central problem in stochastic systems theory: Given a stationary stochastic

process $s_t$, find a Markov process $\tilde{s}_t$, called the state process, and a function $f$ such that

$$s_t = f(\tilde{s}_t) \quad \text{for all } t. \tag{4.2}$$

Moreover, find an Itô stochastic differential equation whose unique solution is the state process $\tilde{s}_t$. The problem of characterizing the family of all such representations is known as the stochastic realization problem [81].

What kind of processes $s_t$ can be expressed as a function of a Markov process (4.2)?

There is a rather comprehensive theory of stochastic realization for the case where $s_t$ is a Gaussian process (which occurs, for example, when $x_t$ is a Gaussian process). This theory expresses $s_t$ as a linear map of an Ornstein–Uhlenbeck process [39,82,83]. The idea is as follows: as a mean-zero Gaussian process, $s_t$ is completely determined by its autocovariance function $C(t - r) = \mathbb{E}[s_t \otimes s_r]$, which by stationarity only depends on $|t - r|$. It is well known that any mean-zero stationary Gaussian process with exponentially decaying autocovariance function is an Ornstein–Uhlenbeck process (a result sometimes known as Doob's theorem) [39,84–86]. Thus if $C$ equals a finite sum of exponentially decaying functions, we can express $s_t$ as a linear function of several nested Ornstein–Uhlenbeck processes, i.e. as an integrator chain from control theory [87,88]

$$
\begin{aligned}
s_t &= f(s_t^{(0)}) \\
\mathrm{d}s_t^{(0)} &= f_0(s_t^{(0)}, s_t^{(1)})\mathrm{d}t + \varsigma_0 \mathrm{d}W_t^{(0)} \\
\mathrm{d}s_t^{(1)} &= f_1(s_t^{(1)}, s_t^{(2)})\mathrm{d}t + \varsigma_1 \mathrm{d}W_t^{(1)} \\
&\vdots \qquad \vdots \qquad \vdots \\
\mathrm{d}s_t^{(n-1)} &= f_{n-1}(s_t^{(n-1)}, s_t^{(n)})\mathrm{d}t + \varsigma_{n-1} \mathrm{d}W_t^{(n-1)} \\
\mathrm{d}s_t^{(n)} &= f_n(s_t^{(n)})\mathrm{d}t + \varsigma_n \mathrm{d}W_t^{(n)}.
\end{aligned}
\tag{4.3}
$$

In this example, $f, f_i$ are suitably chosen linear functions, $\varsigma_i$ are matrices and $W^{(i)}$ are standard Brownian motions. Thus, we can see $s_t$ as the output of a continuous-time hidden Markov model, whose (hidden) states $s_t^{(i)}$ encode its various orders of motion: position, velocity, jerk etc. These are known as generalized coordinates of motion in the Bayesian filtering literature [89–91]. See figure 10.

More generally, the state process $\tilde{s}_t$ and the function $f$ need not be linear, which enables to realize nonlinear, non-Gaussian processes $s_t$ [89,92,93]. Technically, this follows as Ornstein–Uhlenbeck processes are the only stationary Gaussian Markov processes. Note that stochastic realization theory is not as well developed in this general case [81,89,93–95].

### (ii) Stochastic control of integrator chains

Henceforth, we assume that we can express $s_t$ as a function of a Markov process $\tilde{s}_t$ (4.2). Inserting (4.2) into (4.1), we now see that the expected autonomous states minimize how far themselves and $f(\tilde{s}_t)$ are from their target value of zero

$$
(a_t, \mu_t) \quad \text{minimizes } (a, \mu) \mapsto \begin{bmatrix} f(\tilde{s}_t), a, \mu \end{bmatrix} K \begin{bmatrix} f(\tilde{s}_t) \\ a \\ \mu \end{bmatrix}. \tag{4.4}
$$

Furthermore, if the state process $\tilde{s}_t$ can be expressed as an integrator chain, as in (4.3), then we can interpret expected active and internal states as controlling each order of motion $s_t^{(i)}$. For example, if $f$ is linear, these processes control each order of motion $s_t^{(i)}$ towards their target value of zero.

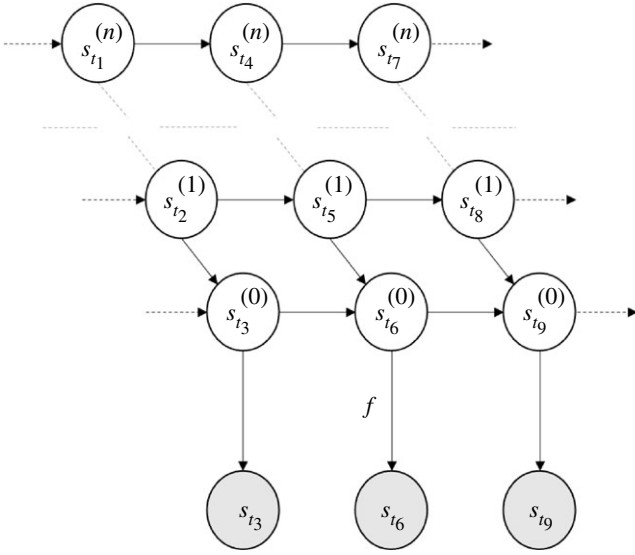

**Figure 10.** Continuous-time Hidden Markov model. This figure depicts (4.3) in a graphical format, as a Bayesian network [3,31]. The encircled variables are random variables—the processes indexed at an arbitrary sequence of subsequent times $t_1 < t_2 < \ldots < t_9$. The arrows represent relationships of causality. In this hidden Markov model, the (hidden) state process $\tilde{s}_t$ is given by an integrator chain—i.e. nested stochastic differential equations $s_t^{(0)}, s_t^{(1)}, \ldots, s_t^{(n)}$. These processes $s_t^{(i)}$, $i \geq 0$, can, respectively, be seen as encoding the position, velocity, jerk etc, of the process $s_t$.

### (iii) PID-like control

PID control is a well-known control method in engineering [77,80]. More than 90% of controllers in engineered systems implement either PID or PI (no derivative) control. The goal of PID control is to control a signal $s_t^{(1)}$, its integral $s_t^{(0)}$, and its derivative $s_t^{(2)}$ close to a pre-specified target value [77].

This turns out to be exactly what happens here when we consider the stochastic control of an integrator chain (4.4) with three orders of motion ($n = 2$). When $f$ is linear, expected autonomous states control integral, proportional and derivative processes $s_t^{(0)}, s_t^{(1)}, s_t^{(2)}$ towards their target value of zero. Furthermore, from $f$ and $K$ one can derive integral, proportional and derivative gains, which penalise deviations of $s_t^{(0)}, s_t^{(1)}, s_t^{(2)}$, respectively, from their target value of zero. Crucially, these control gains are simple by-products of the steady-state density and the stochastic realization problem.

Why restrict ourselves to PID control when stochastic control of integrator chains is available? It turns out that when sensory states $s_t$ are expressed as a function of an integrator chain (4.3), one may get away by controlling an approximation of the true (sensory) process, obtained by truncating high orders of motion as these have less effect on the dynamics, though knowing when this is warranted is a problem in approximation theory. This may explain why integral feedback control ($n = 0$), PI control ($n = 1$) and PID control ($n = 2$) are the most ubiquitous control methods in engineering applications. However, when simulating biological control—usually with highly nonlinear dynamics—it is not uncommon to consider generalized motion to fourth ($n = 4$) or sixth ($n = 6$) order [92,96].

It is worth mentioning that PID control has been shown to be implemented in simple molecular systems and is becoming a popular mechanistic explanation of behaviours such as bacterial chemotaxis and robust homeostatic algorithms in biochemical networks [77,97,98]. We suggest that this kind of behaviour emerges in Markov blankets at non-equilibrium steady state. Indeed, stationarity means that autonomous states will look as if they respond adaptively

to external perturbations to preserve the steady state, and we can identify these dynamics as implementations of various forms of stochastic control (including PID-like control).

## 5. Discussion

In this paper, we considered the consequences of a boundary mediating interactions between states internal and external to a system. On unpacking this notion, we found that the states internal to a Markov blanket look as if they perform variational Bayesian inference, optimizing beliefs about their external counterparts. When subdividing the blanket into sensory and active states, we found that autonomous states perform active inference and various forms of stochastic control (i.e. generalizations of PID control).

### (a) Interacting Markov blankets

The sort of inference we have described could be nuanced by partitioning the external state-space into several systems that are themselves Markov blankets (such as Markov blankets nested at several different scales [1]). From the perspective of internal states, this leads to a more interesting inference problem, with a more complex generative model. It may be that the distinction between the sorts of systems we generally think of as engaging in cognitive, inferential, dynamics [99] and simpler systems rest upon the level of structure of the generative models (i.e. steady-state densities) that describe their inferential dynamics.

### (b) Temporally deep inference

This distinction may speak to a straightforward extension of the treatment on offer, from simply inferring an external state to inferring the trajectories of external states. This may be achieved by representing the external process in terms of its higher orders of motion by solving the stochastic realization problem. By repeating the analysis above, internal states may be seen as inferring the position, velocity, jerk, etc of the external process, consistently with temporally deep inference in the sense of a Bayesian filter [91] (a special case of which is an extended Kalman–Bucy filter [100]).

### (c) Bayesian mechanics in non-Gaussian steady states

The treatment from this paper extends easily to non-Gaussian steady states, in which internal states appear to perform approximate Bayesian inference over external states. Indeed, any arbitrary (smooth) steady-state density may be approximated by a Gaussian density at one of its modes using a so-called Laplace approximation. This Gaussian density affords one with a synchronization map in closed form[4] that maps the expected internal state to an approximation of the expected external state. It follows that the system can be seen as performing approximate Bayesian inference over external states—precisely, an inferential scheme known as variational Laplace [101]. We refer the interested reader to a worked-out example involving two sparsely coupled Lorenz systems [30]. Note that variational Laplace has been proposed as an implementation of various cognitive processes in biological systems [25,52,58] accounting for several features of the brain's functional anatomy and neural message passing [53,70,99,102,103].

### (d) Modelling real systems

The simulations presented here are as simple as possible and are intended to illustrate general principles that apply to all stationary processes with a Markov blanket (3.1). These principles have been used to account for synthetic data arising in more refined (and more specific) simulations of an interacting particle system [27] and synchronization between two sparsely coupled stochastic Lorenz systems [30]. Clearly, an outstanding challenge is to account for empirical data arising

[4]Another option is to empirically fit a synchronization map to data [27].

from more interesting and complex structures. To do this, one would have to collect time-series from an organism's internal states (e.g. neural activity), its surrounding external states, and its interface, including sensory receptors and actuators. Then, one could test for conditional independence between internal, external and blanket states (3.1) [104]. One might then test for the existence of a synchronization map (using lemma 2.3). This speaks to modelling systemic dynamics using stochastic processes with a Markov blanket. For example, one could learn the volatility, solenoidal flow and steady-state density in a stochastic differential equation (3.2) from data, using supervised learning [105].

# 6. Conclusion

This paper outlines some of the key relationships between stationary processes, inference and control. These relationships rest upon partitioning the world into those things that are internal or external to a (statistical) boundary, known as a Markov blanket. When equipped with dynamics, the expected internal states appear to engage in variational inference, while the expected active states appear to be performing active inference and various forms of stochastic control.

The rationale behind these findings is rather simple: if a Markov blanket derives from a steady-state density, the states of the system will look as if they are responding adaptively to external perturbations in order to recover the steady state. Conversely, well-known methods used to build adaptive systems implement the same kind of dynamics, implicitly so that the system maintains a steady state with its environment.

Data accessibility. All data and numerical simulations can be reproduced with code freely available at https://github.com/conorheins/bayesian-mechanics-sdes.

Authors' contributions. Conceptualization: L.D.C., K.F., C.H., G.A.P. Formal analysis: L.D.C., K.F., G.A.P. Software: L.D.C., C.H. Supervision: K.F., G.A.P. Writing-original draft: L.D.C. Writing-review and editing: K.F., C.H., G.A.P. All authors gave final approval for publication and agree to be held accountable for the work performed therein.

Competing interests. We have no competing interests.

Funding. L.D. is supported by the Fonds National de la Recherche, Luxembourg (Project code: 13568875). This publication is based on work partially supported by the EPSRC Centre for Doctoral Training in Mathematics of Random Systems: Analysis, Modelling and Simulation (EP/S023925/1). K.F. was a Wellcome Principal Research Fellow (Ref: 088130/Z/09/Z). C.H. is supported by the U.S. Office of Naval Research (N00014-19-1-2556). The work of G.A.P. was partially funded by the EPSRC, grant no. EP/P031587/1, and by JPMorgan Chase & Co. Any views or opinions expressed herein are solely those of the authors listed, and may differ from the views and opinions expressed by JPMorgan Chase & Co. or its affiliates. This material is not a product of the Research Department of J.P. Morgan Securities LLC. This material does not constitute a solicitation or offer in any jurisdiction.

Acknowledgements. L.D. would like to thank Kai Ueltzhöffer, Toby St Clere Smithe and Thomas Parr for interesting discussions. We are grateful to our two anonymous reviewers for feedback which substantially improved the manuscript.

# Appendix A. Existence of synchronization map: proof

We prove lemma 2.3.

*Proof.* (i) $\iff$ (ii) follows by definition of a function.
(ii) $\iff$ (iii) is as follows

$$\forall b_1, b_2 \in \mathcal{B} : \boldsymbol{\mu}(b_1) = \boldsymbol{\mu}(b_2) \Rightarrow \boldsymbol{\eta}(b_1) = \boldsymbol{\eta}(b_2)$$

$$\iff (\forall b_1, b_2 \in \mathcal{B} : \Sigma_{\mu b} \Sigma_b^{-1} b_1 = \Sigma_{\mu b} \Sigma_b^{-1} b_2 \Rightarrow \Sigma_{\eta b} \Sigma_b^{-1} b_1 = \Sigma_{\eta b} \Sigma_b^{-1} b_2)$$

$$\iff (\forall b \in \mathcal{B} : \Sigma_{\mu b} \Sigma_b^{-1} b = 0 \Rightarrow \Sigma_{\eta b} \Sigma_b^{-1} b = 0)$$

$$\iff \ker \Sigma_{\mu b} \subset \ker \Sigma_{\eta b}$$

(iii) $\iff$ (iv) From [106, §0.7.3], using the Markov blanket condition (2.3), we can verify that

$$\Pi_\mu \Sigma_{\mu b} = -\Pi_{\mu b} \Sigma_b$$

and

$$\Pi_\eta \Sigma_{\eta b} = -\Pi_{\eta b} \Sigma_b.$$

Since $\Pi_\mu, \Pi_\eta, \Sigma_b$ are invertible, we deduce

$$\ker \Sigma_{\mu b} \subset \ker \Sigma_{\eta b}$$
$$\iff \ker \Pi_\mu \Sigma_{\mu b} \subset \ker \Pi_\eta \Sigma_{\eta b}$$
$$\iff \ker -\Pi_{\mu b} \Sigma_b \subset \ker -\Pi_{\eta b} \Sigma_b$$
$$\iff \ker \Pi_{\mu b} \subset \ker \Pi_{\eta b}.$$

$\blacksquare$

# Appendix B. The Helmholtz decomposition

We consider a diffusion process $x_t$ on $\mathbb{R}^d$ satisfying an Itô stochastic differential equation (SDE) [39,107,108],

$$dx_t = f(x_t)dt + \varsigma(x_t)dW_t, \tag{B 1}$$

where $W_t$ is an $m$-dimensional standard Brownian motion (a.k.a., Wiener process) [38,39] and $f : \mathbb{R}^d \to \mathbb{R}^d, \varsigma : \mathbb{R}^d \to \mathbb{R}^{d \times m}$ are smooth functions satisfying for all $x, y \in \mathbb{R}^d$:

— Bounded, linear growth condition: $|f(x)| + |\varsigma(x)| \le K(1 + |x|)$,
— Lipschitz condition: $|f(x) - f(y)| + |\varsigma(x) - \varsigma(y)| \le K|x - y|$,

for some constant $K$ and $|\varsigma| = \sum_{ij} |\varsigma_{ij}|$. These are standard regularity conditions that ensure the existence and uniqueness of a solution to the SDE (B1) [108, theorem 5.2.1].

We now recall an important result from the theory of stationary diffusion processes, known as the Helmholtz decomposition. It consists of splitting the dynamic into time-reversible (i.e. dissipative) and time-irreversible (i.e. conservative) components. The importance of this result in non-equilibrium thermodynamics was originally recognized by Graham in 1977 [109] and has been of great interest in the field since [39,110–112]. Furthermore, the Helmholtz decomposition is widely used in statistical machine learning to generate Monte-Carlo sampling schemes [39,73,113–116].

**Lemma B.1 (Helmholtz decomposition).** *For a diffusion process* (B1) *and a smooth probability density $p > 0$, the following are equivalent:*

(i) *$p$ is a steady state for $x_t$.*
(ii) *We can write the drift as*

$$f = f_{rev} + f_{irrev}$$
$$f_{rev} := \Gamma \nabla \log p + \nabla \cdot \Gamma \tag{B 2}$$
$$f_{irrev} := Q \nabla \log p + \nabla \cdot Q.$$

*where $\Gamma = \varsigma \varsigma^\top / 2$ is the diffusion tensor and $Q = -Q^\top$ is a smooth antisymmetric matrix field. $\nabla \cdot$ denotes the divergence of a matrix field defined as $(\nabla \cdot Q)_i := \sum_j (\partial/\partial x_j) Q_{ij}$.*

*Furthermore, $f_{rev}$ is invariant under time-reversal, while $f_{irrev}$ changes sign under time-reversal.*

In the Helmholtz decomposition of the drift (B2), the diffusion tensor $\Gamma$ mediates the dissipative flow, which flows towards the modes of the steady-state density, but is counteracted by random fluctuations $W_t$, so that the system's distribution remains unchanged—together these

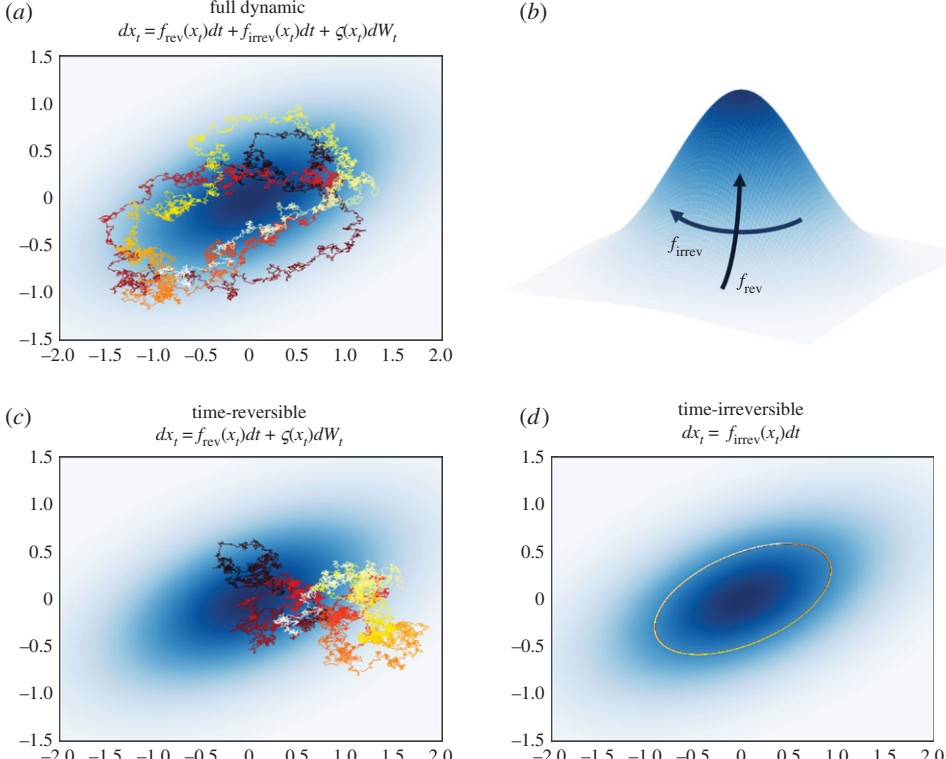

**Figure 11.** Helmholtz decomposition. (*a*) A sample trajectory of a two-dimensional diffusion process (B1) on a heat map of the (Gaussian) steady-state density. (*b*) The Helmholtz decomposition of the drift into time-reversible and time-irreversible parts: the time-reversible part of the drift flows towards the peak of the steady-state density, while the time-irreversible part flows along the contours of the probability distribution. The lower panels plot sample paths of the time-reversible (*c*) and time-irreversible (*d*) parts of the dynamics. Purely conservative dynamics (*d*) are reminiscent of the trajectories of massive bodies (e.g. planets) whose random fluctuations are negligible, as in Newtonian mechanics. The lower panels help illustrate the meaning of time-irreversibility: if we were to reverse time (cf. (B3)), the trajectories the time-reversible process would be, on average, no different, while the trajectories of the time-irreversible process would flow, say, clockwise instead of counterclockwise, which would clearly be distinguishable. Here, the full process (*a*) is a combination of both dynamics. As we can see the time-reversible part affords the stochasticity while the time-irreversible part characterizes non-equilibria and the accompanying wandering behaviour that characterizes life-like systems [11,117]. (Online version in colour.)

form the time-reversible part of the dynamics. In contrast, $Q$ mediates the solenoidal flow—whose direction is reversed under time-reversal—which consists of conservative (i.e. Hamiltonian) dynamics that flow on the level sets of the steady state. See figure 11 for an illustration. Note that the terms time-reversible and time-irreversible are meant in a probabilistic sense, in the sense that time-reversibility denotes invariance under time-reversal. This is opposite to reversible and irreversible in a classical physics sense, which respectively mean energy preserving (i.e. conservative) and entropy creating (i.e. dissipative).

*Proof.* ' $\Rightarrow$ ' It is well known that when $x_t$ is stationary at $p$, its time-reversal is also a diffusion process that solves the following Itô SDE [118]

$$dx_t^- = f^-(x_t^-)dt + \varsigma(x_t^-)dW_t$$
$$f^- := -b + p^{-1}\nabla \cdot (2\Gamma p).$$

(B3)

This enables us to write the drift $f$ as a sum of two terms: one that is invariant under time reversal, another that changes sign under time-reversal

$$f = \frac{f + f^-}{2} + \frac{f - f^-}{2} =: f_{\text{rev}} + f_{\text{irrev}}.$$

It is straightforward to identify the time-reversible term

$$f_{\text{rev}} = p^{-1}\nabla \cdot (\Gamma p) = p^{-1}\Gamma\nabla p + p^{-1}p\nabla \cdot \Gamma = \Gamma\nabla\log p + \nabla \cdot \Gamma. \qquad \text{(B4)}$$

For the remaining term, we first note that the steady-state $p$ solves the stationary Fokker–Planck equation [39,119]

$$\nabla \cdot (-fp + \nabla \cdot (\Gamma p)) = 0.$$

Decomposing the drift into time-reversible and time-irreversible parts, we obtain that the time-irreversible part produces divergence-free (i.e. conservative) flow w.r.t. the steady-state distribution

$$\nabla \cdot (f_{\text{irrev}}p) = 0.$$

Now recall that any (smooth) divergence-free vector field is the divergence of a (smooth) antisymmetric matrix field $A = -A^T$ [109,110,120]

$$f_{\text{irrev}}p = \nabla \cdot A.$$

We define a new antisymmetric matrix field $Q := p^{-1}A$. It follows from the product rule for divergences that we can rewrite the time-irreversible drift as required

$$f_{\text{irrev}} = Q\nabla\log p + \nabla \cdot Q.$$

'⇐' From (B4), we can rewrite the time-reversible part of the drift as

$$f_{\text{rev}} = p^{-1}\nabla \cdot (\Gamma p). \qquad \text{(B5)}$$

In addition, we define the auxiliary antisymmetric matrix field $A := pQ$ and use the product rule for divergences to simplify the expression of the time-irreversible part

$$f_{\text{irrev}} = p^{-1}\nabla \cdot A.$$

Note that

$$\nabla \cdot (f_{\text{irrev}}p) = 0$$

as the matrix field $A$ is smooth and antisymmetric. It follows that the distribution $p$ solves the stationary Fokker–Planck equation

$$\nabla \cdot (-fp + \nabla \cdot (\Gamma p)) = \nabla \cdot (-f_{\text{rev}}p - f_{\text{irrev}}p + \nabla \cdot (\Gamma p)) = \nabla \cdot (-f_{\text{irrev}}p) = 0.$$

∎

## Appendix C. Free energy computations

The free energy reads (3.6)

$$F(b, \mu) = D_{\text{KL}}[q_\mu(\eta)||p(\eta|b)] - \log p(b, \mu).$$

Recalling from (2.3) and (3.3) that $q_\mu(\eta)$ and $p(\eta|b)$ are Gaussian, the KL divergence between multivariate Gaussians is well known

$$q_\mu(\eta) = \mathcal{N}(\eta; \sigma(\mu), \Pi_\eta^{-1}) \quad \text{and} \quad p(\eta|b) = \mathcal{N}(\eta; \boldsymbol{\eta}(b), \Pi_\eta^{-1})$$

$$\Rightarrow D_{\text{KL}}[q_\mu(\eta)||p(\eta|b)] = \frac{1}{2}(\sigma(\mu) - \boldsymbol{\eta}(b))\Pi_\eta(\sigma(\mu) - \boldsymbol{\eta}(b)).$$

Furthermore, we can compute the log partition

$$-\log p(b, \mu) = \frac{1}{2} \left[ b, \mu \right] \Sigma_{b:\mu}^{-1} \begin{bmatrix} b \\ \mu \end{bmatrix} \quad \text{(up to a constant).}$$

Note that $\Sigma_{b:\mu}^{-1}$ is the inverse of a principal submatrix of $\Sigma$, which in general differs from $\Pi_{b:\mu}$, a principal submatrix of $\Pi$. Finally,

$$F(b, \mu) = \frac{1}{2}(\sigma(\mu) - \boldsymbol{\eta}(b))\Pi_\eta(\sigma(\mu) - \boldsymbol{\eta}(b)) + \frac{1}{2} \left[ b, \mu \right] \Sigma_{b:\mu}^{-1} \begin{bmatrix} b \\ \mu \end{bmatrix} \quad \text{(up to a constant).}$$

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
