## [Peer Review File · Proceedings. Mathematical, Physical, and Engineering Sciences]

Review History

RSPA-2021-0518.R0 (Original submission)

Review form: Referee 1

Is the manuscript an original and important contribution to its field?

Excellent

Is the paper of sufficient general interest?

Good

Is the overall quality of the paper suitable?

Good

Can the paper be shortened without overall detriment to the main message?

Yes

Do you think some of the material would be more appropriate as an electronic appendix?

No

Do you have any ethical concerns with this paper?

No

Recommendation?

Accept with minor revision (please list in comments)

Comments to the Author(s)

Please see attached file for review comments. (See Appendix A)

Review form: Referee 2

Is the manuscript an original and important contribution to its field?

Acceptable

Is the paper of sufficient general interest?

Acceptable

Is the overall quality of the paper suitable?

Acceptable

Can the paper be shortened without overall detriment to the main message?

Yes

Do you think some of the material would be more appropriate as an electronic appendix?

No

Do you have any ethical concerns with this paper?

No

Recommendation?

Major revision is needed (please make suggestions in comments)

Comments to the Author(s)

My comments are in the attached document. (See Appendix B)

Decision letter (RSPA-2021-0518.R0)

29-Sep-2021

Dear Mr Da Costa

The Editor of Proceedings A has now received comments from referees on the above paper and would like you to revise it in accordance with their suggestions which can be found below (not including confidential reports to the Editor).

Please submit a copy of your revised paper within four weeks - if we do not hear from you within this time then it will be assumed that the paper has been withdrawn. In exceptional circumstances, extensions may be possible if agreed with the Editorial Office in advance.

Please note that it is the editorial policy of Proceedings A to offer authors one round of revision in which to address changes requested by referees. If the revisions are not considered satisfactory by the Editor, then the paper will be rejected, and not considered further for publication by the journal. In the event that the author chooses not to address a referee's comments, and no scientific justification is included in their cover letter for this omission, it is at the discretion of the Editor whether to continue considering the manuscript.

To revise your manuscript, log into <http://mc.manuscriptcentral.com/prsa> and enter your Author Centre, where you will find your manuscript title listed under "Manuscripts with Decisions." Under "Actions," click on "Create a Revision." Your manuscript number has been appended to denote a revision.

You will be unable to make your revisions on the originally submitted version of the manuscript. Instead, revise your manuscript and upload a new version through your Author Centre.

When submitting your revised manuscript, you will be able to respond to the comments made by the referee(s) and upload a file "Response to Referees" in Step 1: "View and Respond to Decision Letter". Please provide a point-by-point response to the comments raised by the reviewers and the editor(s). A thorough response to these points will help us to assess your revision quickly. You can also upload a 'tracked changes' version either as part of the 'Response to reviews' or as a 'Main document'.

IMPORTANT: Your original files are available to you when you upload your revised manuscript. Please delete any unnecessary previous files before uploading your revised version.

When revising your paper please ensure that it remains under 28 pages long. In addition, any pages over 20 will be subject to a charge (£150 + VAT (where applicable) per page). Your paper has been ESTIMATED to be 26 pages.

Open Access

You are invited to opt for open access, our author pays publishing model. Payment of open access fees will enable your article to be made freely available via the Royal Society website as soon as it is ready for publication. For more information about open access please visit <https://royalsociety.org/journals/authors/open-access/>. The open access fee for this journal is £1700/\$2380/€2040 per article. VAT will be charged where applicable. Please note that if the corresponding author is at an institution that is part of a Read and Publishing deal you are required to select this option. See <https://royalsociety.org/journals/librarians/purchasing/read-and-publish/read-publish-agreements/> for further details.

Once again, thank you for submitting your manuscript to Proc. R. Soc. A and I look forward to receiving your revision. If you have any questions at all, please do not hesitate to get in touch.

Yours sincerely
Raminder Shergill
proceedingsa@royalsociety.org

on behalf of
Dr Marco Mazza
Board Member
Proceedings A

Reviewer(s)' Comments to Author:
Referee: 1
Comments to the Author(s)
Please see attached file for review comments.

Referee: 2
Comments to the Author(s)
My comments are in the attached document.

Board Member:

Comments to Author(s):

Authors should address the suggestions of both referees, and the criticism of the second.

RSPA-2021-0518.R1 (Revision)

Review form: Referee 1

Is the manuscript an original and important contribution to its field?

Excellent

Is the paper of sufficient general interest?

Good

Is the overall quality of the paper suitable?

Excellent

Can the paper be shortened without overall detriment to the main message?

Yes

Do you think some of the material would be more appropriate as an electronic appendix?

No

Do you have any ethical concerns with this paper?

No

Recommendation?

Accept as is

Comments to the Author(s)

Thank you for the updates. I think it's a great paper. Congratulations.

Review form: Referee 2

Is the manuscript an original and important contribution to its field?

Good

Is the paper of sufficient general interest?

Good

Is the overall quality of the paper suitable?

Good

Can the paper be shortened without overall detriment to the main message?

Yes

Do you think some of the material would be more appropriate as an electronic appendix?

No

Do you have any ethical concerns with this paper?

No

Recommendation?

Accept as is

Comments to the Author(s)

I am happy with the changes, modifications and additions made by the author. In my opinion it would be a nice contribution and hence recommend publication of the article.

Decision letter (RSPA-2021-0518.R1)

27-Oct-2021

Dear Mr Da Costa

I am pleased to inform you that your manuscript entitled "Bayesian mechanics for stationary processes" has been accepted in its final form for publication in Proceedings A.

Our Production Office will be in contact with you in due course. You can expect to receive a proof of your article soon. Please contact the office to let us know if you are likely to be away from e-mail in the near future. If you do not notify us and comments are not received within 5 days of sending the proof, we may publish the paper as it stands.

As a reminder, you have provided the following 'Data accessibility statement' (if applicable). Please remember to make any data sets live prior to publication, and update any links as needed when you receive a proof to check. It is good practice to also add data sets to your reference list. Statement (if applicable): All data and numerical simulations can be reproduced with code freely available at [\url{https://github.com/conorheins/bayesian-mechanics-sdes}](https://github.com/conorheins/bayesian-mechanics-sdes)

Open access

You have opted for open access. Payment of open access fees will enable your article to be made freely available via the Royal Society website as soon as it is ready for publication. For more information about open access please visit <https://royalsociety.org/journals/authors/which-journal/open-access/>. The open access fee for this journal is £1700/\$2380/€2040 per article. VAT will be charged where applicable.

Note that if you have opted for open access then payment will be required before the article is published – payment instructions will follow shortly.

Your article has been estimated as being 28 pages long. Our Production Office will inform you of the exact length at the proof stage.

Proceedings A levies charges for articles which exceed 20 printed pages. (based upon approximately 540 words or 2 figures per page). Articles exceeding this limit will incur page charges of £150 per page or part page, plus VAT (where applicable).

Under the terms of our licence to publish you may post the author generated postprint (ie. your accepted version not the final typeset version) of your manuscript at any time and this can be made freely available. Postprints can be deposited on a personal or institutional website, or a recognised server/repository. Please note however, that the reporting of postprints is subject to a

media embargo, and that the status the manuscript should be made clear. Upon publication of the definitive version on the publisher's site, full details and a link should be added.

You can cite the article in advance of publication using its DOI. The DOI will take the form: 10.1098/rspa.XXXX.YYYY, where XXXX and YYYY are the last 8 digits of your manuscript number (eg. if your manuscript number is RSPA-2017-1234 the DOI would be 10.1098/rspa.2017.1234).

For tips on promoting your accepted paper see our blog post:
<https://royalsociety.org/blog/2020/07/promoting-your-latest-paper-and-tracking-your-results/>

On behalf of the Editor of Proceedings A, we look forward to your continued contributions to the Journal.

Sincerely,
Raminder Shergill
proceedingsa@royalsociety.org

on behalf of
Dr Marco Mazza
Board Member
Proceedings A

Reviewer(s)' Comments to Author:

Referee: 1

Comments to the Author(s)

Thank you for the updates. I think it's a great paper. Congratulations.

Referee: 2

Comments to the Author(s)

I am happy with the changes, modifications and additions made by the author. In my opinion it would be a nice contribution and hence recommend publication of the article.

Appendix A

August 8, 2021

Dear authors:

I like to congratulate you with this nice summary of Bayesian mechanics for stationary processes. This paper summarizes the main results of how separation of internal and external states by a Markov blankets leads to some interesting self-organizing dynamics that links directly to other well-known fields, such Bayesian machine learning and control theory. To my knowledge, some, if not most, of these results have been published in related works by the same authors, but it's useful to see all these results together in a short paper.

The paper is concise, which I appreciate, and yet tries to link many fields. I have two main comments:

1. For full comprehension, the reader should be familiar with other published works by the authors and also have good working knowledge of diverse mathematical branches such as probability theory, linear algebra, and stochastic differential equations. This is a heavy demand on the reader and therefore I think most readers would benefit more from a bit more tutorial style.
2. As this is a paper for the "mathematical, physical, and engineering sciences" collection, I think it is fair to expect very clean mathematics in the paper. At times, this aspect can be improved.

Below, I provide some details and examples that resulted in my impression as described above.

- (pg.4, ln.55). Please write either $p(x) = \mathcal{N}(x|0, \Pi^{-1})$ or $x \sim \mathcal{N}(0, \Pi^{-1})$, but not $p(x) = \mathcal{N}(0, \Pi^{-1})$. Please check this throughout the paper, as this notational inaccuracy shows up at more place, e.g., eq. (2.3).
- (pg.4, ln.55). Wrt the same equation, why introduce $p(x) = \mathcal{N}(0, \Pi^{-1})$ rather than $p(x) = \mathcal{N}(0, \Sigma)$? It's the same equation without the inversion, which is just notational overhead that increases cognitive burden on the reader.
- (pg.4, ln.55). What means $\Pi \succ 0$? I guess something like Π is positive definite, but please explain all non-standard symbols.
- (pg.5, ln.8). Suddenly $\Pi_{\mu\eta}$ appears without any explanation of what this is. I guess (after working on the example) it's a cross-precision matrix, but please define its place properly in the matrix Π rather than stating eq. 2.2 without any context, so we can follow the example.
- (pg.5, ln.10). In the example (2.1), $\Sigma_{\eta:b}$ appears and the reader wonders

why the semi-colon is present in $\Sigma_{\eta;b}$ but not in $\Pi_{\mu\eta}$. All my comments are little things, but these little things add up and keeps the reader working hard on figuring out what's intentional and what's just an inaccuracy. I spent about 10 minutes on re-deriving Example 2.1, which, if presented with accurate and complete math could have been less than 2 minutes.

- (pg.5, ln.37). It is not uncommon to denote vectors by bold font, but here bold font is used to indicate expectations. The bold font for expectations leads to very hard to read equations such as pg.11, ln.40. Please use standard notational conventions for math and physics. How about $\bar{\eta} = E[\eta|b]$ or $\hat{\eta} = E[\eta|b]$?
- (pg.5, line 51 and also pg.6 ln 45). These are figures without figure numbers (nor captions).
- (pg.6, ln.26) Nowhere in the text is the word kernel (or image of a matrix) even mentioned. Still, equations of images and kernels of matrices appear at many places.
- (pg.6, ln.33, Example 2.2). These examples look more like lemmas. In particular the 2nd example did not help me at all. For instance, it's not clear to me how choosing $\Pi_{\mu b}$ *at random* guarantees full rank for $\Pi_{\mu b}$. What do you mean by at random?
- (pg.7, ln. 37). 2.1 was not introduced as a Lemma.
- (pg.7, ln. 37, footnote 2). [35] is a big paper. Please refer to an equation or section in the paper.
- (pg.8, ln.24). You write $x_t \sim p$ and just before that in the text you refer to 2.1. I guess you mean $x_t \sim \mathcal{N}(0, \Sigma)$? That reads a bit easier. Are you intentionally restricting yourself to a time series x_t whose samples are drawn independently and from an identical distribution?
- (pg.9, eqs.3.3 and 3.4) This is hard to follow. These equations are just stated without explanation. First, it is interesting to see that the mean of $q_\mu(\eta)$ depends on internal states (by $\sigma(\mu)$), but the variance (Π_η^{-1}) of $q_\mu(\eta)$ is apparently not affected by the internal states. Is this intentional, a general result or justified by any means? I also don't understand eq. 3.4. For instance, the RHS $p(\eta|b)$ is conditioned on b , but b is not present in the LHS $q_\mu(\eta)$, although it is in some complicated way because the statement $q_\mu(\eta) = p(\eta|b)$ seems to imply $\mu = \mu(b)$, which is a function of b . It's just not very clear what's on the table here, and the text does not help.
- (Pg.11, ln.39). I just don't know what's going on here

$$\mathbf{a}_t := \mathbf{a}(s_t) := E_{p(a|s_t)}[a] = \Sigma_{as} \Sigma_s^{-1} s_t$$

This statement includes a double definition, there are 3 different variables for a (\mathbf{a}_t , \mathbf{a} and a). This is just not clear.

- (pg.24 ln 30). reference 57 is incomplete (is it a dissertation?). Also, ref

32 please indicate that it's a dissertation.

There are more of these issues, but I hope the examples above illustrate my experience as a reader. I hope that by sharing my reading experience with you, my feedback is more useful than just a list of issues that I want to see fixed. From my viewpoint, this is a great paper, with lots of fantastic materials, but I spent a lot of time trying to figure out what the equations mean *exactly*, since most of them are not derived, and sometimes they are a bit sloppy. If I would spend a few days full-time, I could probably figure out most of the equations and I am sure this paper would teach me a lot of new things. Unfortunately, I (and likely many other readers) don't have time to spend a few days on a paper, so I hope to read papers with very clean mathematical expressions that are carefully introduced and discussed. I understand that you try to cover a lots of materials in a very concise paper here so the challenge is enormous. I hope my review is helpful to you.

kind regards,

Appendix B

In my opinion, the manuscript is well written in the light of the interdisciplinarity of its contents. It tries to model the interface of two interacting dynamical systems by introducing conditional independence given the interface - referred to as the Markov blanket. Under this condition, they construct a synchronization map between expected states of the two dynamic systems. The existence under some assumptions of such a map is proved and ways of construction illustrated - all under the assumptions of Gaussianity of the steady state distribution of the interface. Finally, this has also been extended by considering specific active states in the interface. Further, they show the learning of the external state given expected internal state can be seen as an inferential approach, similar to MAP estimate or variational inference.

All the above is explained very nicely, but I am not able to judge the novelty of this paper from a pure interdisciplinary perspective. From a statistical point of view, the mathematical derivations, connections and results are rather trivial. Hence I can not comment whether this brings any new directions in biology or applied fields. Broadly speaking this paper reads more like a review paper describing Markov blanket and synchronization map for all different disciplines.

Saying so, I believe a clear illustration (eg. a real data problem and illustration of a match between the proposed setup and reality and applicability of that to learn something?) of what all the tools developed here or proposed are lacking and introduction of this can justify publication of this manuscript in the present venue. This would also help to convey the message of this paper to all the stakeholders it addresses.

Some of the sections, like 4(b) specifically and broadly what is the goal of section 4 is not very clear. It refers and connects to a lot of other techniques from engineering verbally, but where and how all these connections are used is missing. I believe detailed mathematical description and application to some setup is necessary to illustrate what is going on in that section.

There are some notational inconsistency, eg. $\Pi_{b:\mu}$ and $\Pi_{b\mu}$ have been used interchangeably - are they same or different? Please check carefully.